# Generative Neural Machine Translation

**Harshil Shah**[1]
[1]University College London

**David Barber**[1,2,3]
[2]Alan Turing Institute    [3]reinfer.io

## Abstract

We introduce Generative Neural Machine Translation (GNMT), a latent variable architecture which is designed to model the semantics of the source and target sentences. We modify an encoder-decoder translation model by adding a latent variable as a language agnostic representation which is encouraged to learn the meaning of the sentence. GNMT achieves competitive BLEU scores on pure translation tasks, and is superior when there are missing words in the source sentence. We augment the model to facilitate multilingual translation and semi-supervised learning without adding parameters. This framework significantly reduces overfitting when there is limited paired data available, and is effective for translating between pairs of languages not seen during training.

## 1  Introduction

Data from multiple modalities (*e.g.* an image and a caption) can be used to learn representations with more understanding about their environment compared to when only a single modality (an image or a caption alone) is available. Such representations can then be included as components in larger models which may be responsible for several tasks. However, it can often be expensive to acquire multi-modal data even when large amounts of unsupervised data may be available.

In the machine translation context, the same sentence expressed in different languages offers the potential to learn a representation of the sentence's meaning. Variational Neural Machine Translation (VNMT) [Zhang et al., 2016] attempts to achieve this by augmenting a baseline model with a latent variable intended to represent the underlying semantics of the source sentence, achieving higher BLEU scores than the baseline. However, because the latent representation is dependent on the source sentence, it can be argued that it doesn't serve a different purpose to the encoder hidden states of the baseline model. As we demonstrate empirically, this model tends to ignore the latent variable therefore it is not clear that it learns a useful representation of the sentence's meaning.

In this paper, we introduce Generative Neural Machine Translation (GNMT), whose latent variable is more explicitly designed to learn the sentence's semantic meaning. Unlike the majority of neural machine translation models (which model the *conditional* distribution of the target sentence given the source sentence), GNMT models the *joint* distribution of the target sentence and the source sentence. To do this, it uses the latent variable as a language agnostic representation of the sentence, which generates text in both the source and target languages. By giving the latent representation responsibility for generating the same sentence in multiple languages, it is encouraged to learn the semantic meaning of the sentence. We show that GNMT achieves competitive BLEU scores on translation tasks, relies heavily on the latent variable and is particularly effective at translating long sentences. When there are missing words in the source sentence, GNMT is able to use its learned representation to infer what those words may be and produce good translations accordingly.

We then extend GNMT to facilitate multilingual translation whilst sharing parameters across languages. This is achieved by adding two categorical variables to the model in order to indicate the source and target languages respectively. We show that this parameter sharing helps to reduce the impact of overfitting when the amount of available paired data is limited, and proves to be effective

for translating between pairs of languages which were not seen during training. We also show that by setting the source and target languages to the same value, monolingual data can be leveraged to further reduce the impact of overfitting in the limited paired data context, and to provide significant improvements for translating between previously unseen language pairs.

## 2 Model

**Notation** $\mathbf{x}$ denotes the source sentence (with number of words $T_x$) and $\mathbf{y}$ denotes the target sentence (with number of words $T_y$). $\mathbf{e}(v)$ is embedding of word $v$.

GNMT models the *joint* probability of the target sentence and the source sentence *i.e.* $p(\mathbf{x}, \mathbf{y})$ by using a latent variable $\mathbf{z}$ as a language agnostic representation of the sentence. The factorization of the joint distribution is shown in equation (1) and graphically in figure 1.

$$p_\theta(\mathbf{x}, \mathbf{y}, \mathbf{z}) = p(\mathbf{z})p_\theta(\mathbf{x}|\mathbf{z})p_\theta(\mathbf{y}|\mathbf{z}, \mathbf{x}) \tag{1}$$

This architecture means that $\mathbf{z}$ models the commonality between the source and target sentences, which is the semantic meaning. We use a Gaussian prior: $p(\mathbf{z}) = \mathcal{N}(\mathbf{0}, \mathbf{I})$. $\theta$ represents the set of weights of the neural networks that govern the conditional distributions of $\mathbf{x}$ and $\mathbf{y}$.

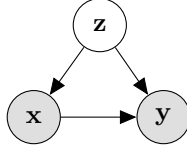

Figure 1: The GNMT graphical model.

### 2.1 Generative process

#### 2.1.1 Source sentence

To compute $p_\theta(\mathbf{x}|\mathbf{z})$, we use a model similar to that presented by Bowman et al. [2016]. The conditional probabilities, for $t = 1, \ldots, T_x$, are:

$$p(x_t = v_x|x_1, \ldots, x_{t-1}, \mathbf{z}) \propto \exp((\mathbf{W}^x \mathbf{h}_t^x) \cdot \mathbf{e}(v_x)) \tag{2}$$

where $\mathbf{h}_t^x$ is computed as:

$$\mathbf{h}_t^x = \text{LSTM}(\mathbf{h}_{t-1}^x, \mathbf{z}, \mathbf{e}(x_{t-1})) \tag{3}$$

#### 2.1.2 Target sentence

To compute $p_\theta(\mathbf{y}|\mathbf{x}, \mathbf{z})$, we modify RNNSearch [Bahdanau et al., 2015] to accommodate the latent variable $\mathbf{z}$. Firstly, the source sentence is encoded using a bidirectional LSTM. The encoder hidden states for $t = 1, \ldots, T_x$ are computed as:

$$\mathbf{h}_t^{\text{enc}} = \overleftrightarrow{\text{LSTM}}(\mathbf{h}_{t\pm1}^{\text{enc}}, \mathbf{z}, \mathbf{e}(x_t)) \tag{4}$$

Then, the conditional probabilities, for $t = 1, \ldots, T_y$, are:

$$p(y_t = v_y|y_1, \ldots, y_{t-1}, \mathbf{x}, \mathbf{z}) \propto \exp((\mathbf{W}^y \mathbf{h}_t^{\text{dec}}) \cdot \mathbf{e}(v_y)) \tag{5}$$

where $\mathbf{h}_t^{\text{dec}}$ is computed as:

$$\mathbf{h}_t^{\text{dec}} = \text{LSTM}(\mathbf{h}_{t-1}^{\text{dec}}, \mathbf{z}, \mathbf{e}(y_{t-1}), \mathbf{c}_t) \tag{6}$$

$$\mathbf{c}_t = \sum_{s=1}^{T_x} \alpha_{s,t} \mathbf{h}_s^{\text{enc}} \tag{7}$$

$$\alpha_{s,t} = \frac{\exp(\mathbf{W}^\alpha [\mathbf{h}_{t-1}^{\text{dec}}, \mathbf{h}_s^{\text{enc}}])}{\sum_{r=1}^{T_x} \exp(\mathbf{W}^\alpha [\mathbf{h}_{t-1}^{\text{dec}}, \mathbf{h}_r^{\text{enc}}])} \tag{8}$$

---

**Algorithm 1** Generating translations

---

Make an initial 'guess' for the target sentence $\mathbf{y}$. This is made randomly, according to a uniform distribution.

**while** not converged **do**

    **E-like step**: Sample $\{\mathbf{z}^{(s)}\}_{s=1}^{S}$ from the variational distribution $q_\phi(\mathbf{z}|\mathbf{x},\mathbf{y})$, where $\mathbf{y}$ is the latest setting of the target sentence.

    **M-like step**: Choose the words in $\mathbf{y}$ to maximize $\frac{1}{S}\sum_{s=1}^{S}\log p(\mathbf{z}^{(s)}) + \log p(\mathbf{x}|\mathbf{z}^{(s)}) + \log p_\theta(\mathbf{y}|\mathbf{z}^{(s)},\mathbf{x})$ using beam search.

**end while**

---

## 2.2 Training

To learn the parameters $\theta$ of the model, we use stochastic gradient variational Bayes (SGVB) to perform approximate maximum likelihood estimation [Kingma and Welling, 2014, Rezende et al., 2014]. To do this, we parameterize a Gaussian inference distribution $q_\phi(\mathbf{z}|\mathbf{x},\mathbf{y}) = \mathcal{N}(\boldsymbol{\mu}_\phi(\mathbf{x},\mathbf{y}), \boldsymbol{\Sigma}_\phi(\mathbf{x},\mathbf{y}))$. This allows us to maximize the following lower bound on the log likelihood:

$$\log p(\mathbf{x},\mathbf{y}) \geq \mathbb{E}_{q_\phi(\mathbf{z}|\mathbf{x},\mathbf{y})}\left[\log\frac{p_\theta(\mathbf{x},\mathbf{y},\mathbf{z})}{q_\phi(\mathbf{z}|\mathbf{x},\mathbf{y})}\right] \equiv \mathcal{L}(\mathbf{x},\mathbf{y}) \tag{9}$$

As per VNMT, for the functions $\boldsymbol{\mu}_\phi(\mathbf{x},\mathbf{y})$ and $\boldsymbol{\Sigma}_\phi(\mathbf{x},\mathbf{y})$, we first encode the source and target sentences using a bidirectional LSTM. For $i \in \{x,y\}$ and $t = 1,\ldots,T_i$:

$$\mathbf{h}_t^{\text{inf},i} = \overleftrightarrow{\text{LSTM}}(\mathbf{h}_{t\pm1}^{\text{inf},i}, \mathbf{e}(i_t)) \quad \text{where } i_t = \begin{cases} x_t & \text{if } i = x \\ y_t & \text{if } i = y \end{cases} \tag{10}$$

We then concatenate the averages of the two sets of hidden states, and use this vector to compute the mean and variance of the Gaussian distribution:

$$\mathbf{h}^{\text{inf}} = \left[\frac{1}{T_x}\sum_{t=1}^{T_x}\mathbf{h}_t^{\text{inf},x}, \frac{1}{T_y}\sum_{t=1}^{T_y}\mathbf{h}_t^{\text{inf},y}\right] \tag{11}$$

$$q_\phi(\mathbf{z}|\mathbf{x},\mathbf{y}) = \mathcal{N}(\mathbf{W}^\mu\mathbf{h}^{\text{inf}}, \text{diag}(\exp(\mathbf{W}^\Sigma\mathbf{h}^{\text{inf}}))) \tag{12}$$

## 2.3 Generating translations

Once the model has been trained (i.e. $\theta$ and $\phi$ are fixed), given a source sentence $\mathbf{x}$, we want to find the target sentence $\mathbf{y}$ which maximizes $p(\mathbf{y}|\mathbf{x}) = \int p_\theta(\mathbf{y}|\mathbf{z},\mathbf{x})p(\mathbf{z}|\mathbf{x})\,d\mathbf{z}$. However, this integral is intractable and so $p(\mathbf{y}|\mathbf{x})$ cannot be easily computed. Instead, because $\arg\max_{\mathbf{y}} p(\mathbf{y}|\mathbf{x}) = \arg\max_{\mathbf{y}} p(\mathbf{x},\mathbf{y})$, we can perform approximate maximization by using a procedure inspired by the EM algorithm [Neal and Hinton, 1998]. We increase a lower bound on $\log p(\mathbf{x},\mathbf{y})$ by iterating between an E-like step and an M-like step, as described in algorithm 1.

## 2.4 Translating with missing words

Unlike architectures which model the conditional probability of the target sentence given the source sentence, $p(\mathbf{y}|\mathbf{x})$, GNMT is naturally suited to performing translation when there are missing words in the source sentence, because it can use the latent representation to infer what those missing words may be.

Given a source sentence with visible words $\mathbf{x}^{\text{vis}}$ and missing words $\mathbf{x}^{\text{miss}}$, we want to find the settings of $\mathbf{x}^{\text{miss}}$ and $\mathbf{y}$ which maximize $p(\mathbf{x}^{\text{miss}},\mathbf{y}|\mathbf{x}^{\text{vis}})$. However, this quantity is intractable as it suffers from a similar issue to that described in section 2.3. Therefore, we use a procedure similar to algorithm 1, increasing a lower bound on $\log p(\mathbf{x}^{\text{vis}},\mathbf{x}^{\text{miss}},\mathbf{y})$, as described in algorithm 2.

## 2.5 Multilingual translation

We extend GNMT to facilitate multilingual translation, referring to this version of the model as GNMT-MULTI. We add two categorical variables to GNMT, $l_x$ and $l_y$ (encoded as one-hot vectors),

---
**Algorithm 2** Translating when there are missing words
---
Make an initial 'guess' for the target sentence $\mathbf{y}$ and the missing words in the source sentence $\mathbf{x}^{\mathrm{miss}}$. These are made randomly, according to a uniform distribution.

**while** not converged **do**

    **E-like step**: Sample $\{\mathbf{z}^{(s)}\}_{s=1}^{S}$ from the variational distribution $q_\phi(\mathbf{z}|\mathbf{x}, \mathbf{y})$, where $\mathbf{x}$ is the latest setting of the source sentence and $\mathbf{y}$ is the latest setting of the target sentence.

    **M-like step (1)**: Choose the missing words in the source sentence $\mathbf{x}^{\mathrm{miss}}$ to maximize $\frac{1}{S}\sum_{s=1}^{S}\log p(\mathbf{z}^{(s)}) + \log p_\theta(\mathbf{x}^{\mathrm{vis}}, \mathbf{x}^{\mathrm{miss}}|\mathbf{z}^{(s)})$ using beam search.

    **M-like step (2)**: Choose the words in $\mathbf{y}$ to maximize $\frac{1}{S}\sum_{s=1}^{S}\log p(\mathbf{z}^{(s)}) + \log p(\mathbf{x}|\mathbf{z}^{(s)}) + \log p_\theta(\mathbf{y}|\mathbf{z}^{(s)}, \mathbf{x})$ using beam search, where $\mathbf{x}$ is the latest setting of the source sentence.

**end while**
---

which indicate what the source and target languages are respectively. The joint distribution is:

$$p_\theta(\mathbf{x}, \mathbf{y}, \mathbf{z}|l_x, l_y) = p(\mathbf{z})p_\theta(\mathbf{x}|\mathbf{z}, l_x)p_\theta(\mathbf{y}|\mathbf{z}, \mathbf{x}, l_x, l_y) \tag{13}$$

This structure allows for parameters to be shared regardless of the input and output languages, and when the amount of available paired translation data is limited, this parameter sharing can significantly mitigate the risk of overfitting. The forms of the neural networks in GNMT-MULTI are identical to those in GNMT (as described in sections 2.1 and 2.2), except that $l_x$ and $l_y$ are now concatenated with the embeddings $\mathbf{e}(x_t)$ and $\mathbf{e}(y_t)$ respectively.

### 2.6 Semi-supervised learning

Monolingual data can be used within the GNMT-MULTI framework to perform semi-supervised learning. This is simply done by setting the source and target language variables $l_x$ and $l_y$ to the same value, in which case the model must attempt to reconstruct the input sentence, rather than translate it. In section 4, we show that when the amount of available paired translation data is limited, using monolingual data in this way further reduces overfitting compared to cross-language parameter sharing alone. Note that we are not concerned about the encoder simply copying the sentence across to the decoder, because the cross-language parameter sharing prevents this.

## 3 Related work

Whilst there have been many attempts at designing generative models of text [Bowman et al., 2016, Dieng et al., 2017, Yang et al., 2017], their usage for translation has been limited. Most closely related to our work is Variational Neural Machine Translation (VNMT) [Zhang et al., 2016], which introduces a latent variable $\mathbf{z}$ with the aim of capturing the source sentence's semantics. It models the conditional probability of the target sentence given the source sentence as $p(\mathbf{y}|\mathbf{x}) = \int p_\theta(\mathbf{y}|\mathbf{z}, \mathbf{x})p_\theta(\mathbf{z}|\mathbf{x})\, d\mathbf{z}$. The authors find that VNMT achieves improvements over modeling $p(\mathbf{y}|\mathbf{x})$ directly (i.e. without a latent variable). The primary difference compared to our work is that VNMT does not model the probability distribution of the source sentence. We believe that learning the joint distribution is a more difficult task than learning the conditional, however this is not without benefit because when learning the joint distribution, the latent variable is more explicitly encouraged to learn the semantic meaning, as shown in the examples in section 4. In addition, because the latent representation is dependent on the source sentence, it is not clear that it serves a different purpose to the encoder hidden states.

Also related is the work of Shu et al. [2017], which presents an approach for using unlabeled data for conditional density estimation. The authors propose a hybrid framework that regularizes the conditionally trained parameters towards the jointly trained parameters. Experiments on image modeling tasks show improvements over conditional training alone.

In work similar to GNMT-MULTI, Johnson et al. [2017] perform multilingual translation whilst sharing parameters by prepending, to the source sentence, a string indicating the target language. Unlike GNMT-MULTI, this approach does not indicate the source language.

There have also been various attempts to leverage monolingual data to improve translation models. Zhang and Zong [2016] use source language monolingual data and Sennrich et al. [2016] use

target language monolingual data to create a synthetic dataset with which to augment the original paired dataset. This is done by passing the monolingual data through a pre-trained translation model (trained using the original paired data), thus creating a new synthetic dataset of paired data. This is combined with the original paired data to create a new, larger dataset which is used to train a new model. In both papers, the authors find that their methods obtain improvements over using paired data alone. However, these procedures do not directly integrate monolingual data into a single, unified model.

## 4 Experiments

In this section we evaluate the effectiveness of GNMT and GNMT-MULTI on the 6 permutations of language pairs between English (EN), Spanish (ES) and French (FR) *i.e.* EN $\rightarrow$ ES, ES $\rightarrow$ EN, EN $\rightarrow$ FR, etc. We also train GNMT-MULTI in a semi-supervised manner, as described in section 2.6, and refer to this as GNMT-MULTI-SSL. We compare the performance of GNMT, GNMT-MULTI, and GNMT-MULTI-SSL against that of VNMT, which we believe to be the most closely related model to our work.

### 4.1 Dataset

We use paired data provided by the Multi UN corpus[1][Tiedemann, 2012]. We train each model with a small, medium and large amount of paired data, corresponding to 40K, 400K and 4M paired sentences respectively. For each language pair, we create validation sets of size 5K and test sets of size 10K paired sentences respectively. For the monolingual data used to train GNMT-MULTI-SSL, we use the News Crawl articles from 2009 to 2012, provided for the WMT'13 translation task. There are 20.9M, 2.7M and 4.5M monolingual sentences for EN, ES and FR respectively.

**Preprocessing** For each language, we convert all characters into lowercase and use a vocabulary of the 20,000 most common words from the paired data, replacing words outside of this vocabulary with an unknown word token. We exclude sentences which have a proportion of unknown words greater than 10% and which are longer than 50 words.

### 4.2 Training

We optimize the ELBO, shown in equation (9), using stochastic gradient ascent. For all models, The latent representation $\mathbf{z}$ has 100 units, each of the RNN hidden states has 1,000 units, and the word embeddings are 300-dimensional. To ensure training is fast, we use only a single sample $\mathbf{z}$ per data point from the variational distribution at each iteration. We perform early stopping by evaluating the ELBO on the validation set every 1,000 iterations. We implement both models in Python, using the Theano [Theano Development Team, 2016] and Lasagne [Dieleman et al., 2015] libraries.

#### 4.2.1 Optimization challenges

The ELBO from equation (9) can be expressed as:

$$\mathcal{L}(\mathbf{x}, \mathbf{y}) = \mathbb{E}_{q_\phi(\mathbf{z}|\mathbf{x},\mathbf{y})} \left[ \log p(\mathbf{x}, \mathbf{y}|\mathbf{z}) \right] - D_{\text{KL}} \left[ q_\phi(\mathbf{z}|\mathbf{x}, \mathbf{y}) \, || \, p(\mathbf{z}) \right] \tag{14}$$

As pointed out by Bowman et al. [2016], when training latent variable language models such as the one described in section 2.1.1, the objective function encourages the model to set $q_\phi(\mathbf{z}|\mathbf{x}, \mathbf{y})$ equal to the prior $p(\mathbf{z})$. As a result, the KL divergence term in equation (14) collapses to 0 and the model ignores the latent variable altogether. To address this, we use the following two techniques:

**KL divergence annealing** We multiply the KL divergence term by a constant weight, which we linearly anneal from 0 to 1 over the first 50,000 iterations of training [Bowman et al., 2016, Sønderby et al., 2016].

Table 1: Test set BLEU scores on pure translation for models trained with varying amounts of paired sentences.

| PAIRED DATA | SYSTEM | EN→ES | ES→EN | EN→FR | FR→EN | ES→FR | FR→ES |
|---|---|---|---|---|---|---|---|
| 40K | VNMT | 12.45 | 12.30 | 12.20 | 12.98 | 12.19 | 13.44 |
| | GNMT | 13.55 | 12.84 | 12.47 | 13.84 | 13.26 | 14.95 |
| | GNMT-MULTI | 16.32 | 15.36 | 15.99 | 16.92 | 16.80 | 18.21 |
| | GNMT-MULTI-SSL | 23.44 | 22.25 | 20.88 | 20.99 | 22.65 | 24.51 |
| 400K | VNMT | 33.27 | 31.96 | 27.71 | 27.69 | 28.76 | 31.22 |
| | GNMT | 33.87 | 32.75 | 28.55 | 28.98 | 29.41 | 31.33 |
| | GNMT-MULTI | 40.08 | 38.56 | 35.55 | 37.28 | 36.31 | 38.68 |
| | GNMT-MULTI-SSL | 43.96 | 41.63 | 37.37 | 39.66 | 38.09 | 40.79 |
| 4M | VNMT | 44.10 | 43.03 | 38.06 | 38.56 | 35.28 | 40.27 |
| | GNMT | 44.52 | 43.83 | 37.97 | 38.44 | 35.96 | 40.55 |
| | GNMT-MULTI | 44.43 | 43.91 | 38.02 | 38.67 | 35.57 | 40.79 |
| | GNMT-MULTI-SSL | 45.94 | 45.08 | 39.41 | 40.69 | 38.97 | 42.05 |

Table 2: Test set KL divergence values ($D_{\mathrm{KL}}\left[q_\phi(\mathbf{z}|\mathbf{x},\mathbf{y}) \,||\, p(\mathbf{z})\right]$) for the model trained with 4M paired sentences, averaged across languages.

| SYSTEM | VNMT | GNMT | GNMT-MULTI | GNMT-MULTI-SSL |
|---|---|---|---|---|
| $D_{\mathrm{KL}}$ | 1.104 | 5.581 | 9.661 | 10.915 |

**Word dropout**    In equation (3), the dependence of the hidden state on the previous word means that the RNN can often afford to ignore the latent variable whilst still maintaining syntactic consistency between words. To prevent this from happening, during training we randomly replace the word being passed to the next RNN hidden state with the unknown word token, as suggested by Bowman et al. [2016]. This is parameterized by a drop rate, which we set to 30%.

Word dropout significantly weakens translation performance for VNMT, therefore we use KL divergence annealing when training both models, but only use word dropout when training GNMT.

## 4.3    Results

### 4.3.1    Translation

The procedure for generating translations using GNMT is described in algorithm 1. For VNMT, the conditional likelihood is $p(\mathbf{y}|\mathbf{x}) = \int p_\theta(\mathbf{y}|\mathbf{z},\mathbf{x})p_\theta(\mathbf{z}|\mathbf{x})\,d\mathbf{z}$. This can be maximized by drawing a set of samples $\{\mathbf{z}^{(s)}\}_{s=1}^S$ from $p_\theta(\mathbf{z}|\mathbf{x})$ and then maximizing $\frac{1}{S}\sum_{s=1}^S p_\theta(\mathbf{y}|\mathbf{z}^{(s)},\mathbf{x})$. This is done approximately, using beam search.

We report results on translation tasks in table 1. When trained with 40K and 400K paired sentences, GNMT has a small advantage over VNMT in terms of BLEU scores across all language pairs. However, both models tend to overfit on these relatively small amounts of paired data. As a result, GNMT-MULTI achieves much higher BLEU scores with both 40K and 400K paired sentences, due to the parameter sharing between languages. Adding monolingual data produces yet another significant increase in BLEU scores. In fact, GNMT-MULTI-SSL trained with only 400K paired sentences achieves performance comparable with GNMT trained with 4M paired sentences. Even with 4M paired sentences, adding monolingual data is helpful, with GNMT-MULTI-SSL outperforming the other models.

In table 2, we report the values of the KL divergence term $D_{\mathrm{KL}}\left[q_\phi(\mathbf{z}|\mathbf{x},\mathbf{y}) \,||\, p(\mathbf{z})\right]$ for the model trained with 4M paired sentences. The higher values for GNMT, GNMT-MULTI and GNMT-MULTI-SSL clearly indicate that these models are placing higher reliance on the latent variable than is VNMT.

**BLEU by sentence length**    It is argued by Tu et al. [2016] that attention based translation models suffer 'coverage' issues, particularly on long sentences, because they do not keep track of the number

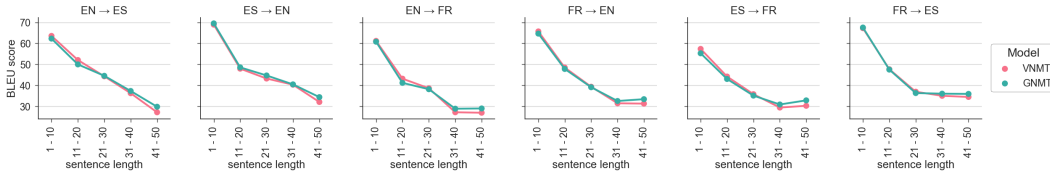

Figure 2: Test set BLEU scores on pure translation, by sentence length, evaluated using the model parameters trained with 4M paired sentences.

Table 3: An example of a long sentence translation, showing the ability of GNMT to capture long range semantics.

| | |
|---|---|
| SOURCE | DANS CE DÉCRET, IL MET EN LUMIÈRE LES PRINCIPALES RÉALISATIONS DE LA RÉPUBLIQUE D'OUZBÉKISTAN DANS LE DOMAINE DE LA PROTECTION ET DE LA PROMOTION DES DROITS DE L'HOMME ET APPROUVE LE PROGRAMME D'ACTIVITÉS MARQUANT LE SOIXANTIÈME ANNIVERSAIRE DE LA DÉCLARATION UNIVERSELLE DES DROITS DE L'HOMME. |
| TARGET | THE DECREE HIGHLIGHTS MAJOR ACHIEVEMENTS BY THE REPUBLIC OF UZBEKISTAN IN THE FIELD OF PROTECTION AND PROMOTION OF HUMAN RIGHTS AND APPROVES THE PROGRAMME OF ACTIVITIES DEVOTED TO THE SIXTIETH ANNIVERSARY OF THE UNIVERSAL DECLARATION OF HUMAN RIGHTS. |
| VNMT | IN THIS REGARD, THE DECREE HIGHLIGHTS THE MAIN ACHIEVEMENTS OF THE UZBEK REPUBLIC IN THE PROMOTION AND PROMOTION AND PROTECTION OF HUMAN RIGHTS AND SUPPORTS THE ACTIVITIES OF THE SIXTIETH ANNIVERSARY OF THE HUMAN RIGHTS OF HUMAN RIGHTS. |
| GNMT | IN THIS DECREE, IT HIGHLIGHTS THE MAIN ACHIEVEMENTS OF THE REPUBLIC OF UZBEKISTAN ON THE PROTECTION AND PROMOTION OF HUMAN RIGHTS AND APPROVES THE ACTIVITIES OF THE SIXTIETH ANNIVERSARY OF THE UNIVERSAL DECLARATION OF HUMAN RIGHTS. |

of times each source word is translated to a target word. However, because the latent variable in GNMT is explicitly encouraged to model the sentence's semantics, it helps to alleviate these issues. This is demonstrated in figure 2 and in the example in table 3, which show that GNMT tends to perform better than VNMT on long sentences, reducing the problems of translating the same phrase multiple times and neglecting to translate others at all.

### 4.3.2 Missing word translation

In order to demonstrate that GNMT does indeed learn a useful representation of the sentence's semantic meaning, we perform missing word translation (*i.e.* where the source sentence has missing words). The model is forced to rely on its learned representation in order to infer what the missing words could be, and then to produce a good translation accordingly.

To produce the missing word data, for each sentence we randomly sample a missing word mask where each word (independently) has a 30% chance of being missing. The procedure for generating translations using GNMT is described in algorithm 2. To generate translations using VNMT, we replace the missing words in the source sentence with the unknown word token and then conduct the same conditional likelihood maximization described in section 4.3.1. The results are reported in table 4. From the BLEU scores, it is evident that GNMT has a significant advantage over VNMT in this scenario, thanks to the quality of its learned representations. We show an example missing word translation in table 5, where the difference in quality between GNMT and VNMT is clear.

### 4.3.3 Unseen language pair translation

Because GNMT-MULTI shares parameters across languages, it should be naturally suited to performing translations between pairs of languages that it never saw during training. For both VNMT and GNMT, to translate, say, from English to Spanish, we first translate from English to French then from French to Spanish (because we assume the English to Spanish parameters are not available). For GNMT-MULTI and GNMT-MULTI-SSL, we train new models where the respective language

Table 4: Test set BLEU scores for missing word translation. We use the model parameters trained with 4M paired sentences.

| SYSTEM | EN → ES | ES → EN | EN → FR | FR → EN | ES → FR | FR → ES |
|--------|---------|---------|---------|---------|---------|---------|
| VNMT   | 26.99   | 27.39   | 23.79   | 23.51   | 22.46   | 25.75   |
| GNMT   | 33.23   | 33.46   | 29.84   | 28.27   | 29.83   | 33.09   |

Table 5: A randomly sampled test set missing word translation from English to Spanish. The struck-through ~~words~~ in the source sentence are considered missing.

| | |
|--------|---|
| SOURCE | WE LOOK ~~FORWARD~~ AT THIS ~~SESSION~~ TO ~~FURTHER~~ MEASURES ~~TO~~ STRENGTHEN THE BEIJING ~~DECLARATION~~ AND PLATFORM ~~FOR~~ ACTION. |
| TARGET | ESPERAMOS QUE EN ESTE PERÍODO DE SESIONES SE ADOPTEN NUEVAS MEDIDAS PARA CONSOLIDAR LA DECLARACIÓN Y LA PLATAFORMA DE ACCIÓN DE BEIJING. |
| VNMT | CONSIDERAMOS QUE EL PERÍODO SE REFIERE A LAS MEDIDAS DE FORTALECIMIENTO DE LA PLATAFORMA DE ACCIÓN DE BEIJING. |
| GNMT | ESPERAMOS CON INTERÉS EN ESTE PERÍODO DE SESIONES UN EXAMEN DE MEDIDAS PARA FORTALECER LA DECLARACIÓN Y LA PLATAFORMA DE ACCIÓN DE BEIJING. |

Table 6: Test set BLEU scores for unseen pair translation. We use the VNMT and GNMT parameters trained with 4M paired sentences. For GNMT-MULTI and GNMT-MULTI-SSL, we train new models with 4M paired sentences, but with the respective language pairs excluded during training.

| | (EN, ES) UNSEEN | | (EN, FR) UNSEEN | | (ES, FR) UNSEEN | |
|---------------|---------|---------|---------|---------|---------|---------|
| SYSTEM        | EN → ES | ES → EN | EN → FR | FR → EN | ES → FR | FR → ES |
| VNMT          | 35.58   | 33.59   | 31.34   | 31.95   | 32.31   | 35.86   |
| GNMT          | 35.35   | 33.76   | 31.55   | 31.38   | 32.39   | 35.85   |
| GNMT-MULTI    | 36.72   | 35.05   | 32.81   | 32.62   | 32.94   | 36.77   |
| GNMT-MULTI-SSL| 38.80   | 37.43   | 34.79   | 34.98   | 33.57   | 38.11   |

pairs are excluded during training. Once trained, we can directly translate from the source to the target language without having to translate to an intermediate language first.

In table 6, we report results on translation between previously unseen language pairs. In this context, VNMT and GNMT perform similarly in terms of BLEU scores. However, both models are consistently outperformed by GNMT-MULTI (albeit only by a small amount). Using monolingual data is very effective in this context, with GNMT-MULTI-SSL outperforming all other models.

## 5 Discussion and future work

In this paper, we have introduced Generative Neural Machine Translation (GNMT), a latent variable architecture which aims to model the semantic meaning of the source and target sentences. For pure translation tasks, GNMT performs competitively with a comparable conditional model, places higher reliance on the latent variable and achieves higher BLEU scores when translating long sentences. When there are missing words in the source sentence, GNMT has superior performance.

We extend GNMT to facilitate multilingual translation without adding parameters to the model. This parameter sharing reduces the impact of overfitting when the amount of available paired data is limited, and proves to be effective for translating between pairs of languages which were not seen during training. We also show that this architecture can be used to leverage monolingual data, which further reduces the impact of overfitting in the limited paired data context, and provides significant improvements for translating between previously unseen language pairs.

Whilst we chose to factorize the joint distribution as per equation (1), this was not the only option we considered. The primary alternative was to use the factorization $p_\theta(\mathbf{x}, \mathbf{y}, \mathbf{z}) = p(\mathbf{z})p_\theta(\mathbf{x}|\mathbf{z})p_\theta(\mathbf{y}|\mathbf{z})$; one could argue that this is in fact more natural for learning sentence semantics, since $\mathbf{z}$ wouldn't

be able to rely on knowing $\mathbf{x}$ explicitly to help generate $\mathbf{y}$ through $p_\theta(\mathbf{y}|\mathbf{x}, \mathbf{z})$. However, experimentally we found that the model struggled to generate grammatically coherent translations which also retained the source sentence's meaning.

We have shown that the idea of using the same sentence in different languages allows for a useful latent representation to be learned. Using these sentence representations could be very promising for use in downstream tasks where 'understanding' of the environment would be helpful, *e.g.* question answering, dialog generation, *etc.*

### Acknowledgments

This work was supported by The Alan Turing Institute under the EPSRC grant EP/N510129/1.

## Footnotes

[1]Whilst the Multi UN corpus forms part of the WMT'13 corpus, we did not use the full WMT'13 corpus since it only provides translations between EN & ES and EN & FR, but not between ES & FR.

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
