[Reviews · NeurIPS 2018]

Reviewer 1



Summary This paper proposes a generative latent variable model for neural machine translation, where inference is performed with variational inference. This extends the work of Zhang et al., 2016, who proposed a conditional model with variational inference. The advantage of the generative model is to force the latent variable to capture more of the semantics of the sentence than the conditional model was able to do. The main disadvantage of this approach is that the value of the latent variable has to be infered during decoding (based on candidate generations). The paper also shows that a version of this model can be trained in a multilingual setting, that monolingual data can be used as semi-supervised training, and that the inference algorithm can be extended to perform translation with missing words. Results show that the model generates better translations than the Variational NMT baseline across multiple language pairs and training data sizes, although the improvement is larger in a small data setting. Multilingual training and semi-supervised training also gives substantial improvements in small data settings. Quality An alternative version of this model would be to condition the inference network q only on x, not on x and y. This would make inference during decoding more efficient, and as z should encode the semantics of either x or y, it is not clear what the additional benefit is of conditioning on both. Experiments: why not also compare directly with previous results on more standard datasets, especially with Zhang et al. 2016's reported results? The paper is also missing a comparison with a standard NMT model without latent variables, without which its performance cannot fully be assessed. Originality The extension over Zhang et al., 2016 is relatively small in terms of the technical contribution, but I do think that the generative formulation has clear advantages over previous work. Significance The main advantage of this approach seems to be in better sampling complexity (more data-efficient learning) than baselines, which is an important direction in machine translation research. More analysis of what semantics the latent vector is learning to capture would be interesting - for example by sampling x and y given z inference on a observed sentence or sentence pair. Overall I think this is an interesting contribution that I'd like to see accepted.

Reviewer 2



This paper describes a latent variable approach to neural machine translation, where the generation of the source and target are conditioned on a latent layer which is claimed to learn the sentence's meaning. The idea is a natural extension of previous work, and also permits a multilingual extension where the source and target word embeddings are annotated with input languages, enabling so-called zero-shot translation and, more interestingly, allowing a more theoretically pleasing integration of monolingual data than prior work that provides a boost in translation quality, as measured by BLEU. It's clear that this line of work is very much at the proof of concept stage, and I think the ideas presented in the paper are novel and interesting. I liked the "missing word" experiments as a means of testing whether the latent annotation learns anything. The authors broach (section 3 paragraph 1) what to me is the most interesting question addressed by this work: how is the latent variable approach, or other similar inductive biases, different from just increasingly the parameterization elsewhere in the model? I think these experiments go a step in the direction of answering that and corroborating the authors claims about what z is doing, but wish there were more experiments in this direction. Also, given how important these experiments are to the paper, it is unfortunate that the paper does not even describe how the deleted data was produced. In general, the paper was very light on these kinds of experimental details that would help with reproducibility. MINOR POINTS Figure 2 is quite hard to read and the resolution provides very little information. Two rows might help. Table 3 could benefit from some annotations calling the reader's attention to important differences and distinctions among the presented hypotheses.

Reviewer 3



Title: Generative Neural Machine Translation Summary: The authors introduce a latent variable into a neural machine translation approach that aims at learning a sentence's semantic meaning. In contrast to common approaches to neural machine translation which model the conditional distribution of the target sentence given the source sentence, GNMT models the joint distribution of the source and target sentence given the latent representation. Because the network's objective is to generate both the source and target sentence from the same latent representation, the latent variable tends to become language agnostic representing the semantic meaning. Strengths: Idea is quite intriguing. Approach facilitates multilingual and zero-shot translations. Weaknesses: Reproducibility: Sample size S is not reported. The test sets were directly sampled (and excluded) from the training data, which makes it fairly easy to yield high BLEU scores. Also, the style of the Multi UN corpus is very specific and somewhat inflates the baseline BLEU scores. To give more support to the suggested method, it would be better to also report BLEU scores on other standard corpora and test sets such as WMT. This would also facilitate a better calibration of the BLEU score results and make them comparable with the state-of-the-art. It might be difficult to find a large enough portion where sentences are fully parallel across more than two languages, though. There is still no guarantee that the model has learned a semantic representation rather than a hidden representation that is similar to what common NMT architectures do. Particularly in the GNMT-Multi setup, it is conceivable that the network learns two separate tasks that form two well separated clusters in the parameter space where one set of parameters is used to do translation and one is used to copy sentences across the decoder. ----------- Thanks to the authors for addressing my comments. Just to clarify my point concerning the two clusters: The capacity of the model may be large enough to actually allow for having two cluster (with potentially some overlap) and not observing any quality loss. A t-SNE visualization might give some clarity here. Anyway, this will be difficult to analyze within the scope of this paper. I definitely would like to see this paper accepted at NIPS, although I still may not place it among the top 50% (yet).